# Meeting demand—Obstetric hemorrhage and blood availability in Malawi, a qualitative study

Stephen E. Njolomole[1,2], Ridhaa Fatima Sachidanandan[3]*, George Mandere[4], Alisa Jenny[5], Adamson S. Muula[6], Bridon M'baya[7], Ben Malinga John[4], Luis Gadama[8], Phylos Bonongwe[9,10], Sylvester Chabunya[1], Evance Storey[10,11], Dilys Walker[12]

1 Department of Pathology, School of Medicine and Oral Health, Kamuzu University of Health Sciences, Blantyre, Malawi, 2 Transfusion Sciences Initiative, Blantyre, Malawi, 3 School of Medicine, University of California, San Francisco, California, United States of America, 4 Department of Population Studies, University of Malawi, Zomba, Malawi, 5 Institute for Global Health Sciences, University of California, San Francisco, California, United States of America, 6 Department of Community and Environmental health, School of Global and Public Health, Kamuzu University of Health Sciences, Blantyre, Malawi, 7 Malawi Blood Transfusion Service, Blantyre, Malawi, 8 Department of Obstetrics and Gynaecology, School of Medicine and Oral Health, Kamuzu University of Health Sciences, Blantyre, Malawi, 9 Department of Obstetrics and Gynaecology, Queen Elizabeth Central Hospital, Blantyre, Malawi, 10 Ministry of Health, Malawi Government, Lilongwe, Malawi, 11 Laboratory Department, Queen Elizabeth Central Hospital, Blantyre, Malawi, 12 Department of Obstetrics, Gynecology and Reproductive Sciences, University of California, San Francisco, California, United States of America

* ridhaa.sachidanandan@ucsf.edu

## Abstract

### Background

Postpartum haemorrhage (PPH) is the leading cause of maternal mortality in Malawi. Despite the presence of a centralized institution supplying blood and blood products for hospitals across the country, a lack of timely blood transfusion has been identified as a critical barrier to successful PPH management. This study aims to understand the factors that affect the blood delivery pipeline and adequate access to blood products for postpartum haemorrhage patients.

### Methods

Qualitative data were collected through in-depth interviews with key stakeholders across the blood delivery pipeline. Interviews were conducted from July 2020 to January 2021 at Queen Elizabeth Central Hospital and Mulanje District Hospital, a referral and district hospital respectively, as well as the Malawi Blood Transfusion Service. Line by line, open coding was used to perform a thematic analysis of the data using Nvivo and Atlas.ti software.

### Results

Five key themes were identified: 1) Lack of blood availability due to an inadequate donor pool, 2) Transportation of blood products and PPH patients is impeded by distance to target sites and competing interests for blood delivery vehicles, 3) The Malawi Blood Transfusion Service has difficulty meeting demand for blood products due to inadequate funding and difficulty retaining blood donors, 4) Current PPH management protocols and practices lead to

**Data Availability Statement:** Due to ethical restrictions, we are unable to share a de-identified data set as the data contain potentially sensitive information that are pertinent to the topic at hand,

thus redaction of said information would result in a loss of context. These restrictions were put in place by the College of Medicine Research and Ethics Committee (COMREC) at the Kamuzu University of Health Sciences (previously known as the University of Malawi, College of Medicine). The research team is happy to provide a minimal anonymized data set on reasonable request, with requests to be sent to COMREC at the following address: IRB Administrator College of Medicine Research and Ethics Committee (COMREC) Kamuzu University of Health Sciences (KUHES) comrecadmin@medcol.mw.

**Funding:** This research was funded by the United States Agency for International Development (USAID) under the Health Evaluation and Applied Research Development (HEARD), Cooperative Agreement No. AID-OAA-A-17-00002. HEARD provided funds to S.E.N. through this award. This study is made possible by the support of the American People through the United States Agency for International Development (USAID). The findings of this study are the sole responsibility of the authors and do not necessarily reflect the views of USAID or the United States Government. The funders had no role in study design, data collection and analysis, decision to publish, or preparation of manuscript. https://urldefense.com/v3/__https://www.heardproject.org/__;!!LQC6Cpwp!s0k-

delays due to inconsistent guidelines on delivery and analysis of patient samples, and 5) Communication between health cadres is inconsistent and affected by a lack of adequate resources.

## Conclusions

Barriers to timely blood transfusion for PPH patients exist across the blood delivery pipeline. While an investment of infrastructure would alleviate many obstacles, several solutions identified in this study can be implemented without additional resources, such as establishing joint department meetings to improve communication between health cadres. Ultimately, given a resource limited setting, it may be worth considering de-centralizing the blood supply.

## Background

Postpartum haemorrhage (PPH) is associated with nearly a quarter of all maternal deaths worldwide making it the leading direct cause of maternal mortality [1]. Despite global progress and interventions, the burden of morbidity and mortality associated with PPH falls primarily on developing nations. According to a 2017 joint analysis between the World Health Organization (WHO) and the United Nations Children's Fund (UNICEF) sub-Saharan Africa is the only region that continues to experience high MMR. Although Malawi's maternal mortality ratio has shown a moderate decline, it remains unacceptably high at 349 per 100,000 live births as of 2017 [2]. Consistent with other countries in sub-Saharan Africa, PPH accounts for at least a quarter of all maternal deaths in Malawi. According to the Malawi National Statistics Office, obstetric haemorrhage remains the most common cause of maternal death [3].

Postpartum haemorrhage is commonly defined as blood loss in excess of 500 mL within 24 hours after vaginal birth (>1000 mL for caesarean delivery), while severe PPH is defined as blood loss of 1000 mL or more within the same timeframe [4]. PPH management can include uterotonic drugs, non-pharmacologic approaches such as uterine tamponade, surgical interventions, and blood transfusion when there is excessive blood loss or when the starting haemoglobin levels were low [4]. When considering the importance of access to blood products, a study conducted in 2002 showed that if a patient requires certain blood products that are not available at the time, the odds of dying are 75 times higher (95% CI 6.98–83.76) [5]. Thus, access to blood within the hospital setting is critical in the management of patients with PPH. This is especially important in Malawi, given that over 90% of births now occur in facilities, compared to less than 30% at the "turn of the millennium" indicating a shift from community to facility births [6].

Prior to 2004, blood was donated at hospitals by family members and friends of patients with PPH, with the potential for exposure to bloodborne illnesses among other safety concerns. In 2004, given the critical need for access to safe blood products, the Malawi Government established the Malawi Blood Transfusion Service (MBTS) in accordance with WHO guidelines to establish a store of blood products under strict measures of quality control [7]. The MBTS is a centralized service responsible for collecting blood from voluntary non-remunerated blood donors and issuing safe blood products to hospitals across Malawi. Hospitals then have the responsibility of collecting blood from the MBTS, cross-matching blood products for specific patients and issuing as needed.

Lack of access to blood products in Malawi may be a critical barrier to successful management of PPH in patients. According to a study conducted in hospitals in the southern region of Malawi, lack of blood transfusion was the main avoidable factor in 18% of maternal deaths, while the Nankumba Safe Motherhood Project found it to contribute to 32% of maternal deaths [8, 9]. Another study done in hospitals in central Malawi found that the main administrative obstacle identified in 20.1% of the maternal death cases studied was a lack of blood availability [10]. While a lack of blood supply has been deemed an issue in PPH cases [11], the specific obstacles towards obtaining blood in facilities with access to blood products have yet to be detailed in a way that can inform interventions to improve care. This qualitative study aims to understand the factors that affect timely and adequate access to blood and blood products for obstetric emergencies in 2 hospitals connected to the MBTS in Malawi.

## Methods

Methods are reported according to the Standards for Reporting Qualitative Research (SRQR) framework [12].

### Study design

This study was a cross-sectional study that used a grounded theory approach to better understand the relationships and behaviors of groups identified as critical to blood supply access and distribution [13]. The study employs a constructivist research paradigm, wherein an understanding of the system came from ongoing discourse with those of relevant lived experience, ultimately resulting in a better-informed consensus of results [14]. Data were collected using In-Depth Interviews (IDIs) with a variety of key stakeholders across the blood supply and delivery chain. The research team was led by principal investigators from Malawi (authors S.E. N., A.S.M.) who have appointments at the Kamuzu University of Health Sciences (formally known as the University of Malawi, College of Medicine) and speak both English and Chichewa. Interviews were conducted by research assistants (RAs) with experience in qualitative data collection, who also speak both English and Chichewa to allow participants to be interviewed in the language of preference. The research assistants did not interact with participants prior to the study. RAs participated in a 5-day training prior to initiating interviews. The training included all aspects of study procedure. A pretest with mock interviews was conducted and informed adjustments to interview guides, which included removing questions that lacked purpose or clarity.

### Study settings

Qualitative data were collected from key informants at three main locations: Queen Elizabeth Central Hospital (QECH), a large urban referral hospital, Mulanje District Hospital (MDH), a rural hospital in Southern Malawi, and MBTS.

QECH, located in Blantyre, is the largest public hospital in Malawi, and serves as a referral location for complex obstetric cases. QECH is one of four regional referral hospitals in Malawi and is representative of the level of care and services provided at these hospitals. The Chatinkha Maternity Unit (CMU), located within QECH, is one of the busiest maternity units in Malawi, reporting over 11,000 deliveries per year. The most recent annual facility data (July 2018-June 2019) lists PPH as the leading cause of maternal deaths at QECH, resulting in 42% of the 463 maternal deaths, out of 11,253 total deliveries [15].

MDH is a rural district hospital located in Mulange, 80 km south of Blantyre, and was selected to represent district hospitals throughout the country which provide relatively limited

services in comparison to regional referral hospitals. Nearly 7,000 live births and 20 maternal deaths were reported at MDH between July 2018—June 2019 [16].

The MBTS, headquarters located in Blantyre, collects blood through clinics at regional offices and mobile clinics for blood donor recruitment campaigns in schools, places of worship and workplaces. Under a mandate instituted by the government on MBTS' inception in 2004, family member donations are only permitted at hospitals when blood is not available at MBTS. In 2019, the MBTS fulfilled approximately 71% of monthly demand at QECH (a monthly average of 959 blood units issued against 1342 requested) and 76% of demand at MDH (monthly average of 144 units issued against 188 requested). In 2020, MBTS fulfilled on average 63% of blood unit demand at QECH (monthly average of 836 units issued against 1322 requested) and 56% of monthly demand at MDH (average of 103 units issued against 183 requested).

The initiation of this study was delayed due to the COVID-19 pandemic and was conducted during the first wave of the virus in Malawi. New coronavirus cases ranged from 33–2573 per month between the months of April 2020 and November 2020, with a total of 6044 new COVID-19 cases during this time period [17].

### Participant sampling

Participants were chosen for this study using purposive sampling to ensure that data was collected from a broad range of stakeholders across the blood supply and delivery spectrum. The sample size was based on the principle of data saturation [18]; while there is no fixed rule regarding sample size for qualitative studies, some authors have recommended a sample of at least 12 participants [18].

The criteria for interview selection, depending on stakeholder group, included working at the respective study site for at least 6 months, recently having worked at the QECH or MDH maternity wards, provided regional supervision to labor and delivery services, or having treated a patient with PPH in the past 6 months that required transfusion. Only those PPH patients who had received a blood transfusion were included in this study. Details regarding the number of IDIs conducted with each cadre and key stakeholder can be found in Table 1.

### Data collection

In-depth interviews were conducted in either English or Chichewa language (the latter is the predominant language spoken in the study area), based on participant preference, from July 2020 to January 2021. The interview guide consisted of a standard set of questions asked across all interviews, with additional follow-up questions asked for clarity and elaboration based on participant response. Subjects were identified and invited to participate by the principal investigators. If interested, the participant agreed upon a date and location for the interview. Written consent was obtained before conducting the interview and all interviews were conducted in private rooms or offices to ensure privacy. IDIs were audio recorded with an electronic recorder and lasted approximately 30–60 minutes based on the content covered. The audio-taped sessions were transcribed by experienced transcribers who also translated the recordings into English. All personal identifiers in the transcripts were removed upon translation. The study leadership team provided ongoing supervision and continuous mentoring throughout the data collection process.

### Data management and analysis

Data were reviewed on an ongoing basis to ensure quality, and feedback was provided to data collectors as needed. Data were stored on an encrypted server at the Kamuzu University of Health Sciences with only investigators having access to password protected files. De-identified

**Table 1. Number of IDIs conducted grouped by study site and cadre.**

| Stakeholder | Number of interviews conducted |
|---|---|
| NATIONAL POLICYMAKERS | 4 |
| Designate of the Parliamentary Committee on Health | 1 |
| Director (or designate) of HTSS | 1 |
| Director (or designate) of Clinical Service | 1 |
| Senior Member of the Malawi Red Cross Society | 1 |
| QECH | 22 |
| Clinicians | 9 |
| Laboratory Technicians | 3 |
| Ward Attendants (Porters) | 2 |
| Laboratory Assistants (Porters) | 2 |
| Drivers | 2 |
| Blood recipients | 3 |
| Family replacement donors | 1 |
| MDH | 14 |
| Clinicians | 8 |
| Ward Attendants | 1 |
| Driver | 1 |
| Blood recipients | 1 |
| Family replacement donors | 3 |
| MBTS | 7 |
| MBTS senior managers | 2 |
| MBTS laboratory employees | 2 |
| Voluntary Non-remunerated blood donors | 3 |
| **Total** | **47** |

data were provided to the research support team at University of California San Francisco (UCSF) through an encrypted data-sharing platform.

A combination of direct and conventional content analysis was employed, in which themes and corresponding codes were theorized in the early stages of data analysis while remaining codes emerged from the data itself [19]. A member of the data analysis team in Malawi (author G.M.) and a member of the research support team at UCSF (author R.F.S.) independently conducted line-by-line open coding of the transcribed interviews [20]. The list of codes and themes were evaluated by the research team throughout the analysis process to ensure that data were being appropriately captured. Data analysis was performed using the NVivo (author G.M.) and ATLAS.ti v8.1 (author R.F.S) software programs [21].

### Ethical considerations and approvals

The study received ethical approval from the College of Medicine Research and Ethics Committee (COMREC) at the University of Malawi (currently the Kamuzu University of Health Sciences) with COMREC approval number P.04 /20 /3037. Participants received a detailed description of the study and provided written consent in advance of the interviews. All participants received a copy of the consent form for their records.

### Results

The data collected in this study were categorized into five overlying themes that impact the provision of blood products to patients with PPH: availability of blood products, transport of

blood products or transfer of patients to target sites, MBTS resources and procedures, hospital/maternity ward clinical policies and procedures for accessing blood products, and communication between health cadres.

## Availability of blood products

Lack of adequate blood supply is the most significant barrier against a timely blood transfusion for PPH patients. Over time, MBTS has increased the number of blood units collected, from an annual total of 4000 units in the year 2004 to approximately 70,000 units in 2018, against an annual target of 120,000 units. Despite this progress, MBTS only managed to meet 60% of the demand for blood products in 2018, and approximately 50% of demand in years prior [22].

Study participants reported that during the study reference period, MDH and QECH experienced shortages of blood and blood products required for transfusion. Many attributed the shortages to the inability of MBTS to collect adequate amounts of blood from voluntary donors. Participants also noted that public understanding of blood donations, based on pre-MBTS systems for collection, has a significant impact on the tendency of the general population to voluntarily donate blood:

> *[People] do refuse completely by saying that they cannot donate blood, asking "why should they donate blood when their relative is not sick?"*

[Voluntary Blood donor].

These findings illustrate the general belief that donating blood is only necessary when a close relative is experiencing blood loss and needs a transfusion. Participants emphasized the need for education regarding the importance of voluntary, non-remunerated blood donation and the significance of having blood stores available for emergency situations. It is especially critical to continuously encourage the general population to donate given the fact that, despite MBTS targeting workplaces, places of worship and other communities as well, the current blood donor population consists primarily of school children. A senior manager at MBTS noted that donations from school blood drives make up to 70% of the institution's blood collections. The impact of this limited diversity of the donor population is important, especially during the onset of the COVID-19 pandemic when many schools closed:

> *When students are on holiday, that becomes a challenge for MBTS to collect enough blood, and especially this year COVID has been quite challenging because most of the students have been at home. So, the availability of blood itself becomes one of the biggest challenge[s] because it's not in adequate* supply

[Ministry of Health Director]

These challenges ultimately lead to a scarcity of blood products that affect the treatment of all patients including those with PPH. Providers noted that oftentimes when ordering blood, the number of units received is less than the number of units requested. Members of the hospital laboratory noted that inadequate stores of blood supply leads to the provision of fewer units than requested. Compounding this issue is the fact that certain blood groups are especially hard to come by:

> *The challenge is that when in the blood bank there is no blood, it is difficult for the patient to receive blood, for example for those who have Rhesus negative, it is very difficult for them to get blood from the blood bank. We had certain case in the labour ward whereby a woman was*

*having PPH, and Rhesus negative and the blood bank did not have the blood and the end result we ended up losing the patient because of scarcity of that blood group*

[Midwife Nurse, CMU, QECH]

Similarly, specific blood products have longer processing times at MBTS and are not always available for patients:

*The other problem is the blood products, you will find that maybe the patient is pancytopenic and you need platelets but you can't find it*

[Medical Officer, QECH]

There is a high demand for blood supply from other departments within the hospitals-e.g. pediatrics and surgery–as well as competition among hospitals for the limited supply available at MBTS. Thus, one of the largest obstacles that PPH patients ultimately face is the availability of this lifesaving product.

## Transport of blood products or transfer of patients to target sites

Transportation of blood to PPH patients is a commonly cited issue by participants in this study. Issues with transportation lie in the availability of vehicles to transport the blood from the MBTS to the facility as well as the distance between MBTS and the rural district hospitals. Blood is collected weekly from MBTS with vehicles being sent from each hospital to pick up blood products at a designated time. This is in contrast to emergency situations, in which a specific blood type or product is requested, and the hospital vehicle must travel to Blantyre and back for the patient to receive transfusion. The distance from Mulanje District Hospital to Blantyre, where MBTS is located, is approximately 65km one way or 1.5 hours in a vehicle. When considering the availability of transport vehicles, the main problem is that these vehicles are also allocated for tasks outside the collection of blood:

*It seems only one vehicle is allocated for daily activities here the same vehicle is shared among several daily activities. This always affects turnaround time*

[QECH, personnel]

For hospitals such as MDH which are located a significant distance away from MBTS, travel time is a significant factor affecting turnaround for blood supply:

*Imagine somebody has PPH in Mulanje but at the laboratory there they don't have blood, they have to find an ambulance to come all over to town (MBTS) to collect blood*

[Voluntary blood donor, College of Medicine / QECH]

Blood product transfusion is not available at all health facilities. If transfusion services are not available, patients need to be transported to facilities such as QECH and MDH. The following quote describes the time it takes to transport patients experiencing PPH. In these cases, there are delays in both the patient and the blood product reaching the facility. In reference to a patient at MDH:

*60kms and returning back is 60kms as well, which is 120kms in total, which can take us almost three hours to go and get the patient and bring her here, because of the structure of our roads,*

*which means she will stay three hours without being assisted, and on top of that, it doesn't mean that the vehicle is just there idle without doing anything, no, and in fact we have enough vehicles here but since they are used for other purposes, so sometimes when they call, informing us that there is a patient, the vehicle maybe is somewhere working, it means it will take almost 5 hours in those far places for that patient to be assisted, so it's like it takes a lot of time*

[Ambulance Driver, MDH]

Transportation of blood units is not only a problem from the MBTS to the hospital but also from the hospital "blood bank" to the maternity ward. In QECH, the Chatinkha Maternity Unit is located on the opposite side of the building from the laboratory. This can affect the drop off time for blood samples that the lab needs for cross-matching, as it takes at least 6 minutes to walk one way from the Chatinkha Maternity Unit to the QECH main lab. This transportation time, alongside a minimum 30-minute wait time for cross matching and issuing of blood, can lead to at least 42 minutes of time elapsed between sample collection and blood unit delivery. Hospital attendants may choose to deliver samples at the end of the workday, as the laboratory is near the exit of the hospital, instead of as they are requested, discussed further in the section on PPH clinical policies and procedures, while others express discomfort at making the journey especially at night:

*I think we have to have our own lab for Chatinkha. . .because its far from here. . .we usually ask the men to go. . . There was this other time when, I think it was my colleague, she went there and then when she was coming back, there was CPR being done, then she came with the blood whilst the patient was already dead.*

[Midwife Nurse, CMU, QECH]

## MBTS resources and procedures

The logistical and administrative burdens of the MBTS impact the provision of blood supply for PPH patients. Many participants noted that lack of funding limits MBTS' ability to reach out to the community with educational campaigns:

*Lack of funding affects our campaigns and now we are even failing to go to our target areas for blood collection campaigns.*

[Laboratory Officer, MBTS]

In addition, adequate funding would allow for the MBTS to purchase the equipment and employ the personnel necessary to process and deliver blood in a more timely manner. The fact that there is only one centralized institution dedicated to blood collection and distribution is another major factor affecting turnaround time:

*We have only one organization which deals with blood donation, which distributes blood to hospitals, so if that organization says it has no blood then we have nowhere else to go to get blood, the end result is that patients die due to lack of blood*

[Midwife Nurse, CMU, QECH]

When discussing the matter of a single institution, participants noted that if the MBTS was able to increase the number of satellite branches with the ability to collect and distribute blood

there would be a significant improvement both on the turnaround time for blood provision–branches in more remote districts will be significantly closer to a source of blood supply–and on the availability of blood itself:

*MBTS is failing to satisfy demand for blood, the demand is more than they can collect but maybe if they extend their branches, they should be found in each and every district in the country, maybe blood collection will increase*

[Nurse Midwife technician, MDH]

Recruiting and retaining voluntary blood donors is another issue noted by study participants, with some saying that the procedures at MBTS regarding treatment of donors are impacting blood donations:

*At times you go there, you meet other people after they have donated blood they say they say they will never go donate blood again because of the way they were treated, I think they should go on a special training on how to welcome or treat blood donors.*

[Voluntary Blood Donor]

## Clinical protocols and practice

While this study primarily focuses on blood access following a diagnosis of PPH, participants discussed the importance of risk assessment of patients to avoid either over burdening or delaying requests:

*Maybe a patient has come with a history of PPH after delivery, they have to follow her up, or any serious condition they may let us know so we can save blood for the patient or she should not deliver, but they already know her history, hence they are supposed to tell us in advance so that we reserve some blood for her and sometimes they send a lot of samples i.e. 20 patients so they can actually grade them in terms of emergencies, priority and waiting list, for us to know that this is an emergency, this is a priority and this one can wait. Yes.*

[Laboratory Technician, QECH]

Many comments implied that updating the current PPH protocols could have an impact on blood delivery. When blood is needed during a PPH case, multiple staff members may be involved. Ward attendants are primarily responsible for dropping patient samples off at the laboratory, however sometimes nurses and ultimately clinicians may be involved in transport of samples and products in an emergency. One of the issues discussed was that sometimes delivery of patient samples to the laboratory is delayed. These delays may result in an even more emergent situation than if the sample had arrived at the lab in a timely manner:

*If they improve their sample delivery to the lab then I think we'll not have any problems with supplying them with blood in good time, because most of the times it's like they are running with the samples from Chatinkha, and Chatinkha is very far from the* lab

[Senior Laboratory Personnel, QECH]

Participants also mentioned how imperative it is that the member of the healthcare team sent to collect the blood waits at the laboratory for the unit so as to not lengthen turnaround

time, although with limited human resources, this may affect care to other patients on the ward:

> *Whosoever comes to collect blood at the lab should wait for us to process blood and give them right away. They should not leave us processing and go somewhere to attend their things, no. and maybe Chatinkha should also call us to say a patient is in need of blood so that you send someone to the lab to collect the pint so this will make us to prepare for the one who is coming to collect blood.*

[Laboratory Technician, QECH]

Similarly, turnaround time can be affected by how rapidly laboratory personnel prepare the blood unit or blood product upon receiving the sample:

> *We are told that we are prioritized here, but sometimes we are also told that right now there is no blood, the only blood that is there is for emergency. When you also question if this is not an emergency what is an emergency?*

[Midwife Nurse, QECH]

This can be further complicated by the delay in hemoglobin drop often seen in the early stages of PPH and the speed with which continued bleeding can compromise an obstetric patient. This can cause delays in blood allocation, as laboratory workers are trained to dispense emergency blood supply in states of low hemoglobin. Such situations often require additional communication between the clinician and laboratory member on duty, further delaying turnaround time. Participants also referenced incomplete blood request forms as a flaw in the PPH response system. In such cases, the laboratory is required to send the form back to the maternity unit to request the additional information prior to dispensing products:

> *I have once worked in the blood bank and I was even the one mostly calling the wards to say "why didn't you fill it completely" because sometimes the patient needs specific volumes, specific blood products, but they don't indicate. There have been times where you process the whole blood unit yet the patient needed only packed cells (Red Cell Suspension).*

[Senior Laboratory Personnel, QECH)

Some participants attributed the issues with incomplete forms to a lack of training, while others specifically noted that the issue may be due to a lack of understanding as to which aspects of the form are critical, especially when it is being filled out in an emergency such as PPH. Ordering of the wrong blood products was also noted as a critical issue that affects the emergency response:

> *Platelets are good because they assist in coagulation. We have had cases where a clinician would order fresh frozen plasma, but the patient is having mucocutaneous bleeding. Us as MBTS, we would advise them that the product you ordered was not ideal, you would have also ordered platelets.*

[MBTS, laboratory personnel]

## Communication between health cadres

The blood delivery pipeline is impacted by miscommunication between health cadres involved in the delivery process. Communication was noted to be most impacted by the lack of functioning phone lines between the maternity department and the laboratory:

> *Okay, there is a problem of phones right now. The phones are not working. In HDU (High dependency unit), since I came, the phones haven't been working. And then, we were using the labor ward phone to phone the lab guys and right now it is not working as well, so it has been a problem whereby at first, the lab guys did not know that our phones are not working, so if we asked someone to go and collect blood or there is an emergency blood sample and we want the results, maybe they would start talking like "You did not phone us" yeah, so, the challenges right now I think are the phones*

[Midwife Nurse, QECH]

With this issue, providers have either had to go to the laboratory in person, a difficult task given its distance and their already busy schedules at the maternity unit or use personal phone credits to contact lab personnel regarding the status of a blood request, to the extent that whether or not a provider has airtime can make a difference in the case. WhatsApp has been used as a backup method of communication however this can lead to further complications, as one provider detailed that they were unable to prepare for a PPH patient since they weren't part of the WhatsApp chain and thus was not informed that a patient was being referred to QECH. This limitation in communication also makes it difficult for the lab team to inform the maternity unit when the blood request is ready, further lengthening turnaround time. Participants also discussed that feedback-related communication between the various departments is hard to come by:

> *The only time that we are communicating is only when we want blood, but there are challenges that they have that they would have liked to actually talk to us maybe formally, or there are challenges that we have but there haven't been meetings whereby we can actually talk about those things. I understand from the lab part, they do hold their meetings or handovers or teaching meetings whatever, of which, a presence of someone from here would actually improve something. Here we do morning meetings. . . but there's no presence of someone from the lab*

[Obstetric Clinical Officer, QECH]

Within the maternity unit, documentation via handover books is key to passing on cases that may be at risk during staff turnover. Poor documentation leads to delays in PPH identification, as key details such as vitals and general appearance may be omitted, in addition to impairing continued treatment plan and diagnosis:

> *I will still go on part of communication, because when we are doing handover cases we have to communicate that there is this particular patient who needs blood transfusion, so the one receiving the handover has to follow up on that case, so if there is no good communication it means that patient will not be attended to*

[Midwife Nurse, QECH]

In the context of accessing blood supply, another crucial line of communication is that between the hospital laboratory and MBTS itself. Advanced communication between the two

departments is key to avoiding communication errors that may lead to mishaps such as family members being sent to donate blood at MBTS despite available blood units, or additional time taken to prepare blood products based on when the transportation department arrives at MBTS and informs them of the request. When discussing turnaround time for blood supply, study participants at MBTS mention that the problem is exacerbated by:

*[A] failure to communicate cause they just come without alerting the lab to say we have an emergency so that can also affect.*

[Laboratory Officer, MBTS]

## Short and long term recommendations

Solutions to the barriers identified in this study were drawn primarily from the interviews conducted with participants, with some input from the authors themselves based on prior experience with MBTS and working as clinicians in the study hospitals (Table 2).

**Table 2. Themes of barriers and challenges and recommendations for interventions to improve timely blood access.**

| Theme | Barrier | Short Term Recommendations | Long Term Recommendations | Stakeholders Involved |
|---|---|---|---|---|
| Availability of blood products | Cultural beliefs and other obstacles resulting in a lack of consistent blood donors | Media campaigns encouraging blood donation | Sensitization campaigns to encourage blood donations Consider allowing family member donations/hospital blood banks | Government, Hospitals (Laboratory) |
| | Scarcity of specific blood groups/ products | Have laboratory reserve blood group O rhesus negative for PPH cases | | |
| Transport of blood products or transfer of patients to target sites | Long distances between MBTS and hospitals leads to increased turnaround time for blood delivery | Assign dedicated vehicles for transport of blood units and patients | Explore alternative transportation models (e.g. drones, motorbikes, delivery of blood by MBTS) | Government, Hospitals |
| | Distance between hospital laboratory and maternity unit leads to increased turnaround time for blood delivery | | Designated lab unit near the maternity center | |
| MBTS resources and procedures | Lack of adequate funding impacts MBTS' ability to conduct blood collection campaigns | | Increased funding for MBTS marketing | Government, MBTS |
| | Lack of a consistent pool of blood donors leads to blood scarcity at MBTS | Implement protocols to improve donor treatment (possibly including improving the provision of food and beverages post donation) | Satellite institutions in other districts Develop protocols and guidelines for decentralized blood collection (including family member donations etc.) | |
| Clinical protocols and practice | Turnaround time for blood products is affected by improper completion of blood request forms and delayed blood sample delivery | Clinical leadership to monitor flow of events Trainings on form completion, pre-natal evaluation of patient and sample delivery | Establishing multi-stakeholder developed PPH emergency protocols | Hospital (Maternity Unit, Laboratory) |
| Communication between health cadres | Resource limitations impact communication between healthcare providers in emergency situations | Install better phone lines in the maternity unit and lab Ensure all clinicians are included on WhatsApp group for emergency communications | | Hospital (Maternity Unit, Laboratory), MBTS |
| | Communication issues between various health departments impacts turnaround time in emergency situations | | Joint department meetings | |

Participants suggested both short term and long term recommendations as solutions to the barriers noted in this study. These included opportunities for more immediate changes such as instructing hospital laboratories to reserve units of O negative blood for obstetric hemorrhage emergencies, as well as longer term strategies to improve supply chain infrastructure and develop campaign strategies to improve supply.

Participants recommend nationwide campaigns that include prominent personalities to encourage the general population to donate, while also clarifying that family member donations are no longer the source of blood transfusions. Emphasis was also placed on the importance of addressing common misconceptions surrounding blood donation and safe donation practices in the time of the COVID-19 pandemic.

With regards to transport of blood and blood products, recommendations included designating a specific vehicle solely for blood product transport in the short-term, while considering other mechanisms of rapid transportation including motorbikes and drones in the long-term. At QECH in particular, the distance between the hospital laboratory and the maternity unit results in safety concerns and time delays that can be alleviated by installing a designated laboratory unit near the maternity ward.

MBTS protocols and procedures affecting blood availability include the ability to retain voluntary blood donors, a process that can be encouraged by implementing protocols that assure appropriate donor treatment and by providing incentives such as food and beverages. Long-term solutions include increased funding for media campaigns, community mobilization and revisiting family member donations until MBTS is able to satisfy the demand for blood products.

To improve clinical protocols around management and treatment of PPH, it was suggested that, in the short term, hospital leadership provide more oversight and trainings on completion of blood request forms, and the delivery of blood samples and blood units. Prenatal risk evaluations could also be performed to screen for women with risk factors for PPH that can prompt pre-emptive blood typing. Additionally, participants recommended that detailed PPH emergency protocols should be developed by stakeholders from all involved hospital departments that can be incorporated into regular trainings for employees.

Communication between departments can be improved by installing functioning phone lines in the hospital while ensuring that all providers are included on WhatsApp threads that are currently being used for communication. Participants recommend that, in the long term, regularly held joint department meetings are crucial for ongoing communication and feedback regarding obstetric hemorrhage management.

## Discussion

Our findings suggest that stakeholders throughout the blood supply and delivery pipeline are aware of its critical importance in treating postpartum hemorrhage, have constructive suggestions, and are keen to implement interventions that can improve timely access to the lifesaving treatment. A variety of barriers involving infrastructure, administrative policies, cultural views, and communication were identified, all of which exacerbate an underlying shortage of blood products.

Blood donation is hindered in the general population due to the belief that the practice is needed only when a family member is at risk, likely reinforced by prior reliance on family members for blood supply before the establishment of the MBTS. Other contributing factors include the long distance from the nearest blood donation site and a lack of time [23]. While changes have been made of late to expand donation campaigns to a larger population such as office workers and among rural communities, the current reliance on school children has

resulted in a seasonal blood supply with certain groups and products more likely to be available when school is in session [24]. This has taken an especially significant toll during the COVID-19 pandemic with the closing of schools, alongside the restrictions on public gatherings, affecting blood campaigns and resulting in a dangerously low blood supply at MBTS (as reflected in the statistics detailing blood supply fulfillment discussed prior). Participants recommend implementing nationwide sensitization and media campaigns that would encourage more people to donate, thus expanding the donor pool beyond school children. Others recommended allowing for family member donations at hospital blood banks for emergency situations [25, 26].

Transportation of blood units and blood-related products–from the MBTS to the hospital as well as from the hospital lab to the maternity ward–and patients experiencing PPH was shown to be hindered both by distance as well as the use of hospital vans for various activities. This finding is corroborated by data identified in a prior study that showed a median travel time to the nearest equipped hospital as approximately 73 minutes with 27% of Malawians living over 120 minutes from an emergency equipped hospital [27]. Within the hospital, specifically at QECH, the distance of the lab from the maternity unit serves as a barrier to delivery of samples and transportation of blood units in emergency situations. Study participants mentioned that having a designated vehicle for blood delivery would be useful in improving turnaround time. Another solution that was frequently recommended was installing a designated lab unit near the maternity unit–a feasible option that is currently in place for the pediatrics department at QECH.

Distance not only plays a factor in Malawians' ability to travel to donate blood, but in MBTS' ability to collect and deliver blood effectively. Without a constant presence in some of the rural districts, it is difficult to retain voluntary non-remunerated blood donors which is key to supporting a centralized blood supply [26]. The national mandate limiting blood collection to MBTS has been praised for its ability to maintain a safe and quality blood supply, however its inability to meet blood demand further contributes to the aforementioned growing body of literature asserting that the Western model of blood banking may be ineffective in a resource limited setting [25, 28]. Lack of funding in a resource limited setting is another factor affecting the success of a centralized blood bank, as advertising campaigns and adequate compensation (in the form of food and beverages for example) that can aid in recruitment and retention of donors require funds beyond that which MBTS is currently provided [25, 26, 28, 29]. Given the logistical difficulties, including funding, of setting up MBTS branches or depots in more districts, participants recommended allowing family blood donors or hospital-based blood banks in order to complement blood supply from MBTS until it is able to satisfy demand (decentralization of blood supply) [25]. Developing policies that decentralize blood supply may be critical in satisfying demand for blood products, a solution that has been emphasized in prior studies [25, 26].

Various aspects of the emergency response protocols and practices were identified as barriers to timely blood access including inaccurate completion of laboratory forms, delayed sample delivery to the lab, lack of prioritization of delivered samples, preparation of blood units in an untimely manner, and delays in picking up prepared blood units. These barriers could be compounded by excessive workloads and insufficient trainings to reinforce emergency response protocols with all cadres in the hospital responsible for blood supply and delivery [30]. Participants recommended identifying the blood type of patients at high risk for developing PPH and providing advanced notice to the hospital laboratory in order to reserve blood products matching the patient's blood type [31]. This is one of many recommendations that could be incorporated into a multi-stakeholder developed PPH protocol, including details regarding form completion, transport, sample prioritization and delivery, and so forth.

Communication barriers between health cadres included both technological and behavioral complications. Resource limitations proved to be a significant barrier to improving the system. For example, a lack of functioning phone lines in the maternity and laboratory units was often cited by participants as an impediment to advanced and ongoing communication between the hospital ward, laboratory and MBTS. Behavioral gaps in communication noted by participants included improper documentation affecting handover of cases at shift change, a lack of advanced communication during emergencies, and minimal feedback between departments regarding workflow. The growing popularity of WhatsApp and other mobile messaging services poses a promising solution to technological communication issues in resource limited settings, however issues such as whether or not providers are included on relevant threads and 'mobile literacy' may serve as challenges to its efficacy [32]. Participants emphasized the importance of joint meetings between the maternity unit and hospital laboratory in order to encourage ongoing communication and feedback, rather than only communicating during a stressful emergency response.

Though this study was specific to Malawi, recent studies have identified similar challenges that impede availability and distribution of blood and blood products for lifesaving transfusions [33, 34]. The findings presented here can thus be applied to similarly resource-limited settings and provide a framework for identifying context specific barriers and solutions to best address the needs of obstetric patients being treated within said system.

## Limitations

This study faced some limitations. First, there is possible response bias in interviews based on subjects' role in blood supply and delivery. Similarly, the use of purposive sampling, while critical for its role in finding knowledgeable participants, may have resulted in sampling bias. MDH was chosen as a study site due to its location and PPH burden and may not be representative of other district hospitals. Additionally, an MBTS member is a co-investigator for this study, thus providing an additional potential source of bias. This was addressed through independent data collection and data analysis conducted by the University of Malawi/Kamuzu University of Health Sciences and UCSF researchers. Finally, study data were collected during the COVID-19 pandemic, which has had a significant impact on blood supply worldwide and likely exacerbated, or possibly even introduced, obstacles identified within the blood delivery pipeline.

## Conclusion

Our study identifies a variety of barriers throughout the blood supply and delivery pipeline that affect timely care for postpartum hemorrhage patients. These are resulting in shortages now more than ever in the face of the COVID-19 pandemic. While many of these obstacles could be resolved by investment in infrastructure, many of the proposed solutions could be implemented without significant additional resources, and established task forces are currently responding using data from this study. Furthermore, until the MBTS has adequate resources to sustainably increase the blood supply across the country, stakeholders may need to consider a decentralized alternative for blood supply collection and distribution.

## Supporting information

**S1 Checklist.**
(DOCX)

## Acknowledgments

We would like to thank all of the stakeholders involved in this study: the Ministry of Health, the Kamuzu University of Health Sciences, Dr Samson Mndolo, the Hospital Director and the entire management and staff at QECH, Dr Patrick Kamalo and all members of the QECH Research Committee, Mrs Natasha Nsamala, management and staff at MBTS, Dr Njikho (the District Medical Officer at MDH), Dr Alinafe Kalanga (Director of Health and Social Services, Mulanje District Council), and the management and staff at MDH

We would also like to acknowledge the following individuals for their various contributions to the study:

Final Lodzani, Isabel Sikwese, Angella Jambo, Misheck Kagalu, Dr, Chimwemwe Nkhonjera, Vincent Mgode, Thom Mfune, Gift Msowoya, Josephine Chimoyo, James Aman, Emmie Malonda, Wongani Nyondo, J Chintsanya, Dr Linda Nyondo Mipando, and Danielle Charlet.

## Author Contributions

**Conceptualization:** Stephen E. Njolomole, Alisa Jenny, Adamson S. Muula, Bridon M'baya, Dilys Walker.

**Data curation:** George Mandere.

**Formal analysis:** Ridhaa Fatima Sachidanandan, George Mandere.

**Funding acquisition:** Stephen E. Njolomole, Adamson S. Muula.

**Investigation:** Stephen E. Njolomole, Adamson S. Muula, Bridon M'baya, Ben Malinga John, Phylos Bonongwe, Sylvester Chabunya, Evance Storey.

**Methodology:** Stephen E. Njolomole, Alisa Jenny, Adamson S. Muula, Bridon M'baya, Ben Malinga John, Luis Gadama, Phylos Bonongwe, Sylvester Chabunya, Evance Storey, Dilys Walker.

**Project administration:** Stephen E. Njolomole.

**Resources:** Stephen E. Njolomole, Alisa Jenny.

**Supervision:** Stephen E. Njolomole.

**Visualization:** Stephen E. Njolomole, Ridhaa Fatima Sachidanandan.

**Writing – original draft:** Stephen E. Njolomole, Ridhaa Fatima Sachidanandan, George Mandere.

**Writing – review & editing:** Stephen E. Njolomole, Ridhaa Fatima Sachidanandan, Alisa Jenny, Adamson S. Muula, Bridon M'baya, Ben Malinga John, Luis Gadama, Phylos Bonongwe, Sylvester Chabunya, Evance Storey, Dilys Walker.

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
