## [Decision Letter · Decision Letter 0]

12 Jun 2022

PONE-D-22-06573Meeting demand - obstetric hemorrhage and blood availability in Malawi, a qualitative studyPLOS ONE

Dear Dr. Sachidanandan,

Thank you for submitting your manuscript to PLOS ONE. After careful consideration, we feel that it has merit but does not fully meet PLOS ONE’s publication criteria as it currently stands. Therefore, we invite you to submit a revised version of the manuscript that addresses the points raised during the review process.

We look forward to receiving your revised manuscript.

Kind regards,

Dylan A Mordaunt, MD, MPH, FRACP

Academic Editor

PLOS ONE

Journal Requirements:

Additional Editor Comments (if provided):

Thank you for your submission. It's a well written piece of work with a clear focus, results and conclusions. All comments are written with the intent of maximising the study quality and addressing the minor issues, with the intent of accepting upon resubmission. With specific regards to the criteria for publication:

1. The study appears to present the results of original research. I agree with the reviewer that citing prior work from other jurisdictions, or systematic reviews, would be helpful where available.

2. Results do not appear to have been published elsewhere.

3. Experiments, statistics, and other analyses are performed to a high technical standard and are described in sufficient detail.

4. Conclusions are presented in an appropriate fashion and are supported by the data.

5. The article is presented in an intelligible fashion and is written in standard English.

6. The authors should clarify whether their institutional review board has granted permission to identify participants by their position description. This would appear to make these people reidentifiable, which may be allowed, but I think the journal would want to clarify this.

7. The article adheres to appropriate reporting guidelines and community standards for data availability. In saying that, I think the study report could benefit from checking compliance with an appropriate reporting checklist such as COREQ or SRQR. I acknowledge that this may appear to push the report towards a more positivist epistemology, but as you'll see in the two cited, that isn't the case and they simply help to ensure all relevant items are reported and available to the wide range of readers PLoS One has. On that note, it would be helpful to clarify whether the authors follow a specific epistemology and method, underlying their methodology. If the authors do use a checklist, it is useful to submit this as a supplementary document. If you have notes generated from analysis, these could also be submitted as supplementary files.

Reviewers' comments:

Reviewer's Responses to Questions

**Comments to the Author**

1. Is the manuscript technically sound, and do the data support the conclusions?

Reviewer #1: Yes

2. Has the statistical analysis been performed appropriately and rigorously? 

Reviewer #1: Yes

3. Have the authors made all data underlying the findings in their manuscript fully available?

Reviewer #1: Yes

4. Is the manuscript presented in an intelligible fashion and written in standard English?

Reviewer #1: Yes

5. Review Comments to the Author

Reviewer #1: Comments

The manuscripts reads well and has a scientific merit to address one of the major causes of maternal mortality in resource poor settings. However, I have some minor comments to improve the manuscript.

Background

Page 2(line 42): Authors might want to search for (and add) some recent systematic analyses, if available

Methods

Page 4 (line 91) : Have you given the abbreviation for SN and AM somewhere before ? If not, better to give the abbreviation here.

Page 5 (line 96) : How was the interview guide revised based on the pretest? What were the major adjustments done on interview guide based on the pre-test results?

Page 5 (lines 101 - 103) : Recheck for the punctuations. I think the 'comma' is required after QECH. The line should go like - Queen Elizabeth Central Hospital (QECH), a large urban referral hospital; Mulanje……………………………………………………..

Page 8 (line 153) : How was the interview guide standardized?

Page 9 (line 183) : Was it the written consent, better to specify.

Results

Overall: Result section focuses on describing the barriers and challenges encountered around five different themes. The solutions/recommendations are given in bullets in table only, not described adequately. I think term used in the title- 'meeting demand' indicates the need of equal weightage to the solution part also. Therefore, I suggest some more elaboration on solutions to barriers/ challenges under each theme.

Page 10 (line 201): What are the societal norms hindering blood donation? The description given below this line specify the individual preferences whether to donate blood or not, but it is not capturing the barriers related to societal norms as a whole. Authors might want to recheck or rethink on use of term 'societal norms' here/ or some descriptions on 'societal norms' can be added (if data are available).

Discussion -

Reads very well.

I would prefer adding one paragraph on strengths, and generalizability of findings in other settings.

6. PLOS authors have the option to publish the peer review history of their article (what does this mean?). If published, this will include your full peer review and any attached files.

Reviewer #1: **Yes: **Dipak Raj Chaulagain

---

## [Author Response · Author response to Decision Letter 0]

6 Aug 2022

Journal Requirements: 

1. We have revised our manuscript to meet PLOS ONE’s style requirements. 

2. Due to ethical restrictions, we are unable to share a de-identified data set as the data contains potentially sensitive information that is pertinent to the topic at hand, thus redaction of said information would result in a loss of context. These restrictions were put in place by the College of Medicine Research and Ethics Committee (COMREC) at the Kamuzu University of Health Sciences (previously known as the University of Malawi, College of Medicine). The research team is happy to provide a minimally anonymized data set on reasonable request, with requests to be sent to COMREC at the following address: 

IRB Administrator 

College of Medicine Research and Ethics Committee (COMREC)

Kamuzu University of Health Sciences (KUHES)

comrecadmin@medcol.mw

Editor/Reviewer Comments:

(Note: All of the revisions and specific line numbers referenced below are based on the “Revised Manuscript with Track Changes” document for efficiency of review)

1. The study appears to present the results of original research. I agree with the reviewer that citing prior work from other jurisdictions, or systematic reviews, would be helpful where available.

Thank you for this suggestion. We have added additional citations, where available, that provide more situational context. 

2. Results do not appear to have been published elsewhere.

Confirming that we have not published these results elsewhere. 

3. Experiments, statistics, and other analyses are performed to a high technical standard and are described in sufficient detail. Thank you. 

4. Conclusions are presented in an appropriate fashion and are supported by the data. Thank you. 

5. The article is presented in an intelligible fashion and is written in standard English. Thank you.

6. The authors should clarify whether their institutional review board has granted permission to identify participants by their position description. This would appear to make these people identifiable, which may be allowed, but I think the journal would want to clarify this.

Thank you for bringing this critical point to our attention. We can confirm that the institutional review board has granted permission to identify participants by their position description, with a few exceptions for individuals whose job descriptions make these people identifiable. These job descriptions were revised accordingly (lines 246-247, 382). 

7. The article adheres to appropriate reporting guidelines and community standards for data availability. In saying that, I think the study report could benefit from checking compliance with an appropriate reporting checklist such as COREQ or SRQR. 

Thank you for suggesting SRQR. We did indeed utilize this checklist as we drafted the manuscript. We have included the checklist in the supplemental materials (note: page and line numbers in the checklist correspond to the final, unmarked “Manuscript” document).

Background

Page 2(line 42): Authors might want to search for (and add) some recent systematic analyses, if available

Thank you for this suggestion. We have revised the document to reference the most recent systematic analyses available.

Methods

Page 4 (line 91) : Have you given the abbreviation for SN and AM somewhere before ? If not, better to give the abbreviation here.

Thank you for this suggestion. We have clarified that these abbreviations refer to author initials in the parentheses (line 111). 

Page 5 (line 96) : How was the interview guide revised based on the pretest? What were the major adjustments done on interview guide based on the pre-test results?

Thank you for this question. The major adjustments done to the interview guide based on pre-test results include the removal of questions that lacked clarity and resulted in responses that did not contribute to the data in a meaningful way. This has been further clarified in line 118-119. 

Page 5 (lines 101 - 103) : Recheck for the punctuations. I think the 'comma' is required after QECH. The line should go like - Queen Elizabeth Central Hospital (QECH), a large urban referral hospital; Mulanje……………………………………………………..

Thank you for this suggestion. We have edited this line (now 123-125) to include proper punctuations.

Page 8 (line 153) : How was the interview guide standardized?

Thank you for this question. The interview guide included a fixed set of questions that were asked of all participating subjects. Additional follow up questions were asked for clarity and further elaboration as needed. This has been clarified in lines 176-178. 

Page 9 (line 183) : Was it the written consent, better to specify.

Thank you for this suggestion, specified in line 210.

Results

Overall: Result section focuses on describing the barriers and challenges encountered around five different themes. The solutions/recommendations are given in bullets in table only, not described adequately. I think term used in the title- 'meeting demand' indicates the need of equal weightage to the solution part also. Therefore, I suggest some more elaboration on solutions to barriers/ challenges under each theme.

Thank you for calling our attention to the discrepancy between what is reported in the table and in the text. We agree that the solutions deserve more description and have added more detail in the text regarding the recommended solutions in lines 441-481. 

Page 10 (line 201): What are the societal norms hindering blood donation? The description given below this line specify the individual preferences whether to donate blood or not, but it is not capturing the barriers related to societal norms as a whole. Authors might want to recheck or rethink on use of term 'societal norms' here/ or some descriptions on 'societal norms' can be added (if data are available).

Thank you for this question. In this case, rather than societal norms per say, it is that the general population is still operating under the principles of the pre-existing methods of blood donation and collection, that is the principle of family member donations at the hospital. This was acknowledged by several subjects as a barrier to voluntary blood donations, as many noted that people often state that they would only need to donate if a family member is in need. The language has been modified to better reflect this concept in lines 229-232. 

Discussion -

Reads very well.

I would prefer adding one paragraph on strengths, and generalizability of findings in other settings.

We have added a few lines (559-563) regarding the generalizability of findings to the end of the discussion section. In this paragraph, we referenced studies that have also sought to assess challenges with and opportunities to improve access and distribution of blood products in other settings.

---

## [Editor Report · Decision Letter 1]

9 Aug 2022

Meeting demand - obstetric hemorrhage and blood availability in Malawi, a qualitative study

PONE-D-22-06573R1

Dear Dr. Sachidanandan,

We’re pleased to inform you that your manuscript has been judged scientifically suitable for publication and will be formally accepted for publication once it meets all outstanding technical requirements.

Kind regards,

Dylan A Mordaunt, MD, MPH, FRACP

Academic Editor

PLOS ONE

Additional Editor Comments (optional):

Thank you for your resubmission. This now meets the criteria for publication.
---

## [Editor Report · Acceptance letter]

15 Aug 2022

PONE-D-22-06573R1 

Meeting demand - obstetric hemorrhage and blood availability in Malawi, a qualitative study 

Dear Dr. Sachidanandan:

I'm pleased to inform you that your manuscript has been deemed suitable for publication in PLOS ONE. Congratulations! Your manuscript is now with our production department. 

Kind regards, 

on behalf of

Associate Professor Dylan A Mordaunt 

Academic Editor

PLOS ONE